# Bioprospection of the Antarctic Diatoms *Craspedostauros ineffabilis* IMA082A and *Craspedostauros zucchelli* IMA088A

**DOI:** 10.3390/md22010035

**Published:** 2024-01-06

**Authors:** Riccardo Trentin, Emanuela Moschin, Luísa Custódio, Isabella Moro

**Affiliations:** 1Department of Biology, University of Padova, Via U. Bassi 58/B, 35131 Padova, Italy; 2Centre of Marine Sciences, Faculty of Sciences and Technology, University of Algarve, Ed. 7, Campus of Gambelas, 8005-139 Faro, Portugal

**Keywords:** Antarctica, biological activity, antioxidant activity, PUFAs, blue biotechnology, extreme environments

## Abstract

In extreme environments such as Antarctica, a diverse range of organisms, including diatoms, serve as essential reservoirs of distinctive bioactive compounds with significant implications in pharmaceutical, cosmeceutical, nutraceutical, and biotechnological fields. This is the case of the new species *Craspedostauros ineffabilis* IMA082A and *Craspedostauros zucchellii* IMA088A Trentin, Moschin, Lopes, Custódio and Moro (Bacillariophyta) that are here explored for the first time for possible biotechnological applications. For this purpose, a bioprospection approach was applied by preparing organic extracts (acetone and methanol) from freeze-dried biomass followed by the evaluation of their in vitro antioxidant properties and inhibitory activities on enzymes related with Alzheimer’s disease (acetylcholinesterase: AChE, butyrylcholinesterase: BChE), Type 2 diabetes mellitus (T2DM, α–glucosidase, α–amylase), obesity (lipase) and hyperpigmentation (tyrosinase). Extracts were then profiled by ultra-high-performance liquid chromatography–mass spectrometry (UPLC–HR–MS/MS), while the fatty acid methyl ester (FAME) profiles were established by gas chromatography–mass spectrometry (GC–MS). Our results highlighted strong copper chelating activity of the acetone extract from *C. ineffabilis* and moderate to high inhibitory activities on AChE, BChE, α–amylase and lipase for extracts from both species. The results of the chemical analysis indicated polyunsaturated fatty acids (PUFA) and their derivatives as the possible compounds responsible for the observed activities. The FAME profile showed saturated fatty acids (SFA) as the main group and methyl palmitoleate (C16:1) as the predominant FAME in both species. Overall, our results suggest both Antarctic strains as potential sources of interesting molecules with industrial applications. Further studies aiming to investigate unidentified metabolites and to maximize growth yield and natural compound production are required.

## 1. Introduction

Recent advances in molecular technologies and the increasing knowledge on the biological diversity of remotes regions on Earth, have rendered bioprospection a convenient and little-disruptive alternative in the exploitation of different environments [1,2,3,4]. Marine bioprospecting aims to draw on the large arsenal of molecules, enzymes and genes present in little-known organisms [1]. Particularly, the study of microalgae and seaweeds revealed already the presence of structurally unique secondary metabolites, new genes, and enzymes with possible commercial value [5,6]. It is estimated that around 15,000 novel compounds were isolated and structurally identified from algae, including lipids, proteins, pigments, carbohydrates, and other chemically active metabolites [5,6,7,8]. Algae are considered as a sustainable and renewable feedstock with significant industrial potential [8,9]. As a result, the algal products market was valued at USD 975.63 million in 2022, and it is expected to grow up to USD 1540.38 million by 2030 [10]. However, while model algae, such as *Chlamydomonas reinhardtii* Dangeard and *Phaeodactylum tricornutum* Bohlin, have been studied for decades and are currently cultivated on an industrial scale [11], little is known regarding non-model extremophilic phototrophs [12]. Particularly, Antarctic algae have evolved unique strategies for surviving in harsh environmental conditions of low temperatures, repeated freeze and thawing cycles, osmotic stress, desiccation, low nutrients availability, variable solar irradiance, and high UV radiation by producing a wide variety of natural products with different structural, and functional properties [12]. Extreme environments, such as Antarctica, are important sources of novel active compounds, which are potentially useful for pharmaceutical, cosmeceutical, nutraceutical, and biotechnological applications [4]. Over the last decade, increasing taxon sampling in Antarctica together with the employment of molecular data resulted in the description of many novel lineages of phototrophs belonging to different taxonomic groups [13,14,15,16]. Among them, two novel species of diatoms of the genus *Craspedostuaros* E.J. Cox, were isolated from sea ice in the Ross Sea, cultivated, sequenced, and described as *Craspedostauros ineffabilis* IMA082A Trentin, Moschin, Lopes, Custódio and Moro and *Craspedostauros zucchellii* IMA088A Trentin, Moschin, Lopes, Custódio and Moro [14]. A preliminary analysis of acetone extracts from both species revealed differences in their metabolic fingerprints, when cultivated at the same growth conditions, suggesting the employment of different survival strategies [14]. The rapid growth of both strains and the promising results in their biochemical profiling, rendered *C. ineffabilis* and *C. zucchellii* as valuable candidates for bioprospecting [14]. For these reasons, in the present work we appraised both species as potential sources of bioactive molecules. With this in mind, both strains were cultivated at the same conditions, and their biomasses were harvested and freeze-dried. Acetone and methanol extracts were prepared from dried biomass and evaluated for in vitro antioxidant activity by radical and metal-based assays and tested as enzymatic inhibitors of acetylcholinesterase (AChE) and butyrylcholinesterase (BChE), both involved in neurological disorders, α–amylase and α–glucosidase, carbohydrate-hydrolyzing enzymes involved in Type 2 diabetes mellitus (T2DM), lipase, implicated in obesity and hyperlipidemia, and tyrosinase, involved in skin hyperpigmentation and food browning. Extracts were also profiled through ultra-high-performance liquid chromatography–mass spectrometry (UPLC–HR–MS/MS) and annotated by using publicly available mass spectral libraries together with an in silico approach. Finally, lyophilized biomass was evaluated in terms of fatty acid methyl ester (FAME) profiles through gas chromatography–mass spectrometry (GC–MS).

## 2. Results and Discussion

### 2.1. In Vitro Antioxidant Properties

In this work, acetone, and methanol extracts from *C. ineffabilis* IMA082A and *C. zucchelli* IMA088A were tested for radical scavenging activity (RSA) against 2,2′-azinobis (3-ethylbenzothiazoline-6-sulfonic acid) (ABTS•+) and 1,1-diphenyl-2-picrylhydrazyl (DPPH•) radicals, for total antioxidant capacity (FRAP) and for metal chelating activities on copper and iron (Figure 1, Table 1 and Appendix A). The acetone extract from *C. ineffabilis* showed significantly higher radical scavenging activity against ABTS•+ and DPPH• compared to the other extracts (Figure 1a,b). The FRAP was similar in terms of acetone extracts from both species and methanol extract from *C. ineffabilis*, whereas the *C. zucchelli* methanol extract showed significantly lower reducing capacity (Figure 1c). Significantly higher copper chelating properties were observed in acetone extracts from *C. ineffabilis* (Figure 1d), whereas other extracts showed slightly lower chelating activities, as reported in ABTS and DPPH assays. Statistically higher iron chelating capacities were reported for both *C. ineffabilis* extracts, with activities above 80% (Figure 1e). For samples reaching over 50% of activity at 10 mg/mL, such as both acetone and methanol extracts for ICA and acetone extract for ABTS from *C. ineffabilis*, the half maximal effective concentrations (EC_50_, mg/mL) were determined (Table 1).

### 2.2. Enzymatic Inhibitory Properties of the Extracts

The inhibitory effects of acetone and methanol extracts of *C. ineffabilis* IMA082A and *C. zucchelli* IMA088A were tested against AChE, BChE, α–amylase, α–glucosidase, lipase and tyrosinase (Figure 2, Table 2 and Appendix A). The extracts showed moderate capability to inhibit AChE and BChE, with methanol extract from both *C. ineffabilis* and *C. zucchelli* and acetone extract from *C. ineffabilis* showing around 60% of inhibition on AChE (Figure 2a). Acetone extract from *C. zucchelli* showed significantly lower inhibitory activity when compared to previous extracts. Acetone extract from *C. ineffabilis* displayed the highest inhibitory capacity against BChE, followed by both acetone and methanol extracts from *C. zucchelli.* The methanol extract from *C. ineffabilis* showed significant lower inhibitory activity (Figure 2b). Acetone extracts from both species showed significantly higher inhibitory capacity on α–amylase and α–glucosidase than the methanol extracts, which displayed nil to low levels of activity (Figure 2c,d). Lipase inhibition was higher in acetone extracts of *C. zucchelli*, followed by acetone extracts of *C. ineffabilis* and methanol extracts from both species (Figure 2e). Statistical differences were observed among methanol and acetone extracts from both species in tyrosinase inhibition, with methanol extracts showing higher inhibitory capacities (Figure 2f). For samples reaching over 50% of inhibition at 10 mg/mL, the half maximal inhibitory concentrations (IC_50_, mg/mL) were determined (Table 2).

### 2.3. Chemical Profile

The chemical profiles of acetone and methanol extracts of *C. ineffabilis* IMA082A and *C. zucchelli* IMA088A were determined through UHPLC–HR–MS/MS. In acetone extracts, 31 peaks showed library spectral matches congruent with in silico analysis (Table 3); thus, these were annotated as ‘putatively annotated compounds’ (level 2), while a total of 9 peaks were annotated in methanol extracts (Table 4). Most of the annotated compounds were lipid-like molecules, such as fatty acids, glycerolipids and oxylipins.

### 2.4. FAMEs Profile

The FAME profiles of *C. ineffabilis* IMA082A and *C. zucchelli* IMA088A were analyzed by GC/MS (Figure 3). Saturated fatty acids (SFA) constituted 31.42 ± 0.11% and 38.54 ± 0.58% of the total detected FAMEs in *C. ineffabilis* and *C. zucchelli*, respectively (Appendix A). Monounsaturated fatty acids (MUFA) represented 36.32 ± 0.37% of the total FAME in *C. ineffabilis* and 30.86 ± 0.85% in *C. zucchelli*, while polyunsaturated fatty acids (PUFA) reported in *C. ineffabilis* were 32.28 ± 0.29% and 30.60 ± 0.55% in *C. zucchelli* (Appendix A). In total, twelve and eleven FAMEs were identified in *C. zucchelli* and *C. ineffabilis*, respectively. Methyl palmitoleate (C16:1) was the predominant FAME in both *C. ineffabilis* (33.60 ± 0.26%) and *C. zucchelli* (24.97 ± 1.40%), followed by methyl palmitate (C16:0) in *C. zucchelli* (23.64 ± 0.45%) and cis–5,8,11,14–eicosatetraenoic methyl ester (C20:4n–6) in *C. ineffabilis* (22.30 ± 0.39%), as reported in Appendix A and Figure 3. Overall, the FAMEs profiles of both diatoms presented the same FAMEs with exception of methyl behenate (C22:0), identified only in *C. zucchelli*. Despite their similar profiles, significant differences were reported among *C. ineffabilis* IMA082A and *C. zucchelli* IMA088A for most of the identified FAMEs. Methyl oleate (or its isomers methyl elaidate, C18:1) and cis–4,7,10,13,16,19–docosahexaenoate (C22:6n–3) were the sole FAMEs showing no significant differences among the two Antarctic species.

### 2.5. Fucoxanthin Content

Fucoxanthin concentrations (Table 5) were spectrophotometrically measured in acetone and methanol extracts of *C. ineffabilis* IMA082A and *C. zucchelli* IMA088A. Methanol extracts from both species showed the highest fucoxanthin content, while acetone extracts exhibited significantly lower concentrations of fucoxanthin (Table 3).

### 2.6. Discussion

Antarctic diatoms have adapted to harsh environmental conditions by developing a wide range of strategies to cope with extreme stressors (e.g., temperature, irradiance, and salinity) [17,18]. Particularly, their physiological and biochemical adaptations might represent the result of unknown evolutionary trajectories leading to the production of molecules with possible ecological, taxonomical, and biotechnological relevance [12,17,18]. This study represents an initial step to test this hypothesis by exploring the bioactive properties and the chemical profile of the Antarctic diatoms *C. ineffabilis* and *C. zucchelli* from Terra Nova Bay (Ross Sea). In a bioprospection effort, organic extracts (acetone and methanol) with different polarities were evaluated for radical scavenging activity (RSA), total antioxidant capacity, metal chelating activities and for in vitro inhibition of enzymes related with human disorders. Although methanol and acetone have similar polarities, the former has a typically polar hydroxyl group and a methyl group, while the latter has one carbonyl group and two methyl groups. Therefore, acetone can extract both polar and nonpolar substances, while methanol targets more polar compounds. The extracts exhibited moderate to low RSA towards ABTS and DPPH radicals with the acetone extract from *C. ineffabilis* showing significantly higher RSA. Generally, all extracts showed limited ability to reduce Fe^3+^ to Fe^2+^, which served as a measure of the total antioxidant activity of the extracts. Similarly, all extracts had moderate ability to chelate Cu^2+^ ions, while *C. ineffabilis* displayed strong Fe^2+^ chelating capacity for both acetone and methanol extracts. The results of in vitro antioxidant properties of the extracts showed overall modest ROS scavenging capacity of *C. ineffabilis* and *C. zucchelli*. The antioxidant activity of the extracts was likely related to the presence of chlorophylls derivatives (e.g., 10S–hydroxypheophorbide a) [19,20,21] detected in the acetone extracts and PUFAs, such as alpha-linolenic acid and eicosapentaenoic acid [22]. PUFAs might be the main responsibilities for the antioxidant activities reported in this study, since lipids and especially PUFAs are known as radical scavengers in microalgae [22,23]. It is known that environmental stressors are key factors that affect the growth performance and the accumulation of valuable compounds [24]. Thus, we expect that non-optimal growth conditions will drive enzymatic and non-enzymatic antioxidant responses to prevent the negative effects of ROS [25,26]. Further analyses are required to evaluate biochemical and physiological responses to stress of *C. ineffabilis* and *C. zucchelli*. Loliolide, a monoterpene common to several marine algae, is a well-known antioxidant, thus its presence in acetone and methanol extracts may explain their antioxidant capacities [27,28].

Acetone and methanol extracts of *C. ineffabilis* IMA082A and *C. zucchelli* IMA088A were further explored as enzymatic inhibitors for pharmaceutical and cosmetic industries. Cholinesterase enzymes (AChE and BChE) catalyze the breaking down of acetylcholine and other choline esters working as neurotransmitters [29]. Thus, the inhibition of AChE and BChE, leading to an increase in neurotransmitters levels, is considered a therapeutic strategy to alleviate symptoms associated with neurodegeneration [30,31]. The methanol extract from both species and the acetone extract from *C. ineffabilis* exhibited high AChE inhibition activity (above 60%), while *C. zucchelli* acetone extract was significantly less effective. BChE inhibition was higher in the acetone extract from *C. ineffabilis*, followed by both acetone and methanol extracts from *C. zucchelli*. It has been recently demonstrated that fucoxanthin, a marine carotenoid common in diatoms, such as *Phaeodactylum tricornutum* [32], and brown seaweeds, such as *Sargassum horneri* (Turner) Agardh [33], displays strong activities against cholinesterase enzymes [33], suggesting the possibility for future development of fucoxanthin as a pharmaceutical or a nutraceutical treatment for neurodegenerative disorders [34]. Fucoxanthin content was determined spectrophotometrically in both *Craspedostauros* species, suggesting the possible role of this carotenoid as AChE and BChE inhibitor. Yet, the chemical analysis of algal and plant extracts suggested a relationship between PUFA content and the inhibition of AChE and BChE [22], particularly α–linolenic acid that has a moderate capacity towards both enzymes [35]. Thus, PUFAs detected in acetone extracts, might act as AChE and BChE inhibitors. Our results showed that both *Craspedostauros* species are potential source of cholinesterase enzymes inhibitors, a trait reported in other microalgae like *Chlorella minutissima* Fott and Nováková, *Tetraselmis chuii* Butcher and *Rhodomonas salina* (Wisłouch) Hill and Wetherbee [22]. Further analyses focusing on carotenoids are necessary to validate the outcomes of this initial screening. In the management of Type 2 diabetes mellitus (T2DM), inhibiting carbohydrate-hydrolyzing enzymes (α-glucosidase and α-amylase) to restrict carbohydrate digestion and glucose absorption plays a fundamental role [29,36]. Marine algae, such as *Gelidiella acerosa* (Forsskål) Feldmann and Hamel, have shown efficacy in inhibiting these enzymes [37]. In this work, only the acetone extract from *C. ineffabilis* had relevant activity towards α–amylase, with an IC_50_ value of 6.87 ± 0.17 mg/mL (Table 2). Eicosapentaenoic acid, a PUFAs annotated by LC–MS analyses, was considered a good α–glucosidase inhibitor (IC_50_ value, 0.10 mM) by Liu et al. [38]; on the contrary, Leporini et al. [39] reported a higher IC_50_ value (0.250 mM) compared to acarbose (IC_50_ value, 0.053 mM), suggesting the low inhibitory activity of this compound. Further studies are required to unravel the role of eicosapentaenoic acid as α–glucosidase inhibitor. Pancreatic lipase catalyzes the hydrolysis of triglyceride into monoglyceride and free fatty acids, representing a preliminary step for their uptake [40]. Thus, the inhibition of this enzyme is crucial in the management of obesity and hyperlipidemia [29]. Acetone extract from *C. zucchelli* showed moderate (41.86 ± 4.58%) lipase inhibition at a concentration of 10 mg/mL and was the extract with the highest inhibitory capacity. Fucoxanthin and fucoxanthinol from the seaweed *Undaria pinnatifida* (Harvey) Suringar showed in vitro inhibition of lipase, suggesting that fucoxanthin from both Antarctic strains might be responsible for the observed activities [41]. Despite orlistat, a common pharmaceutical used in clinical practice, showed higher inhibitory activity then both our extracts (Appendix A), its use is associated with side effects, such as diarrhea. Therefore, crude extracts from *C. ineffabilis* and *C. zucchelli* might be a milder alternative to prevent postprandial hyperlipidemia [41]. Tyrosinase is an enzyme involved in the synthesis of melanin in animals [42] and in the oxidation of phenolics in food [29,43]. Tyrosinase inhibitors are desirable molecules for the treatment of hyperpigmentation and melasma within the cosmetics and medicinal industries [43]. Additionally, they serve as anti-browning agents in the food and agricultural sectors [44]. Methanol extracts from both species showed low tyrosinase inhibition, while acetone extracts showed almost nil inhibitory capacity. Little is known about tyrosinase inhibitors from diatoms, however some specific algal compounds, such as phloroglucinol and its derivatives, showed strong inhibition of tyrosinase activity in the brown alga *Ecklonia stolonifera* [45]. Other interesting biological activities were reported for glycolipids and oxylipins, whose presence was reported acetone extracts from both strains [46,47]. These complex and poorly studied lipids are considered potential anti-tumoral and anti-inflammatory, anti-bacterial, anti-fungal and anti-parasitic compound, thus they could contribute to the plethora of activities reported in this study [46,47]. Further research is mandatory to clarify the biological actions of these molecules.

The lipid fraction of *C. ineffabilis* IMA082A and *C. zucchelli* IMA088A was rich in PUFAs, which constituted approximatively 30% of the total FAMEs in both species. Arachidonic acid methyl ester (C20:4n–6) was the most abundant PUFAs, followed by α–linolenic acids (C18:3) and linoleic acids (C18:2) methyl esters. PUFAs synthesis at cold temperature represents a well-known strategy for the maintenance of membrane fluidity in microalgae [48,49,50]. Our results are consistent with those reported for the Antarctic diatom *Navicula* UMACC 231, which showed around 30% of PUFAs when cultivated at 4 °C [50]. Similarly, higher percentages of PUFAs were reported at lower growth temperature in the green microalga *Chlamydomonas* UMACC 229 from Antarctica [50] and in other eight cold-adapted microalgal strains of different genera (*Chlamydomonas*, *Chlorella*, *Tetraselmis*, *Pseudopleurochloris*, *Nannochloropsis* and *Phaeodactylum*) [51]. The abundance of PUFAs and the presence of highly valuable fatty acids, such as docosahexaenoic acid methyl ester (22:6n–3), render *C. ineffabilis* IMA082A and *C. zucchelli* IMA088A potential sources of these essential nutrients promoting human health and valuable as aquaculture feed [52].

## 3. Materials and Methods

### 3.1. Chemicals

The compounds 2,2′-azinobis (3-ethylbenzothiazoline-6-sulfonic acid) (ABTS•+), 1,1-diphenyl-2-picrylhydrazyl (DPPH•), fatty acid methyl ester (FAME) standards (Supelco^®^ 37 Component FAME Mix) and enzymes were purchased from Sigma (Steinheim am Albuch, Germany). Additional reagents and solvents were obtained from VWR International (Leuven, Belgium).

### 3.2. Biomass Collection and Preparation of the Extracts

*C. ineffabilis* IMA082A and *C. zucchelli* IMA088A were cultivated in F/2 [53] growth medium at a salinity of 35‰, at a temperature of 5 °C and a light intensity of 10 μmol photons m^−2^ × s^−1^. Diatom biomass was harvested at the exponential growth phase by centrifugation at 10,000× *g* for 10 min at 4 °C, supernatant was removed, and the remaining pellet was lyophilized for 24 h. Freeze-dried biomass was stored at room temperature (RT, approx. 20 °C) in the dark for subsequent analyses. Aqueous acetone and aqueous methanol extracts were prepared by mixing freeze-dried biomass (50 mg) of *C. ineffabilis* IMA082A and *C. zucchelli* IMA088A, respectively, with 20 mL of 80% (*v*/*v*) acetone and 20 mL of 50% (*v*/*v*) methanol. The diatom cell walls were disrupted with glass beads in a MM400 mixer mill (Retsch, Haan, Germany) at 30 Hz for 5 min. After 12 h of incubation at 4 °C, samples were centrifuged at 12,000× *g* for 10 min at 4 °C, the supernatants were collected and dried with nitrogen flux overnight. Dried extracts were first diluted at the concentration of 20 mg/mL (stock solutions) and then at 10 mg/mL (working solutions) in the corresponding solvents for the determination of in vitro antioxidant and enzymatic activities. Furthermore, dried extracts were diluted to the concentration of 2 mg/mL and filtered (0.2 nm) for the determination of their chemical profiles through UPLC–HR–MS/MS. For the chemical profiling, a quality control (QC) sample was prepared by mixing equal volumes of each filtered extract.

### 3.3. In Vitro Antioxidant Properties

#### 3.3.1. RSA on DPPH• Radical

The RSA against DPPH• was evaluated by the method of Brand Williams et al. [54], adapted to 96-well microplates by Custódio et al. [55]. Briefly, the extracts (22 µL) were mixed with 200 µL of an ethanol DPPH• solution (120 µM) in 96-well microplates and incubated for 30 min at RT in the darkness. The absorbance was measured at 515 nm. BHT (1 mg/mL) was used as a positive control.

#### 3.3.2. RSA on ABTS•+ Radical

The RSA against ABTS•+ was determined according to the method described by Re et al. [56]. A stock solution of ABTS•+ (7.4 mM) was prepared in ethanol and potassium persulfate (2.6 mM), with an overnight incubation in darkness at 4 °C. The stock solution was then diluted with ethanol to obtain a final absorbance of 0.7 at 734 nm. For the assay, the extracts (10 µL) were mixed with ABTS•+ (190 µL) in 96-well microplates and incubated in darkness at RT for 6 min. The absorbance was measured at 734 nm. BHT (1 mg/mL) was used as a positive control.

#### 3.3.3. FRAP

FRAP was evaluated using the method described by Megías et al. [57]. Extracts (50 µL) were mixed in 96-well microplates with 50 µL of potassium ferricyanide (1% in water) and 50 µL of distilled water. After 20 min of incubation in the darkness at 50 °C, 50 µL of trichloroacetic acid (TCA, 10% in water) and 10 µL of ferric chloride solution (0.1% in water) were added. The absorbance was measured at 700 nm after 10 min of incubation at RT, and BHT was used as the standard.

#### 3.3.4. CCA

CCA was assessed following Megías et al. [57]. Extracts (30 µL) were mixed in 96-well microplates with 200 µL of 50 mM sodium acetate buffer (pH 6), 6 µL of pyrocatechol violet (PV, 4 mM in the acetate buffer) and 100 µL of copper sulphate (50 μg/mL in water). The absorbance was measured at 632 nm. Ethylenediamine tetraacetic acid (EDTA 1 mg/mL) was used as a positive control.

#### 3.3.5. ICA

ICA was determined according to Megías et al. [57]. Extracts (30 μL) were mixed in 96-well microplates with 200 µL of distilled water and 30 µL of an iron (II) chloride solution (0.1 mg/mL in water) and incubated for 30 min at RT. Afterwards, 12.5 µL of ferrozine solution (40 mM in water) was added and the absorbance was measured at 562 nm. EDTA (1 mg/mL) was used as a positive control.

### 3.4. Enzyme Inhibition Assays

#### 3.4.1. AChE and BChE Inhibition

The inhibitory capacity of the extracts on AChE and BChE was evaluated by the method described by Ellman et al. 1961 [58] and adapted to 96 well microplates [59]. In brief, extracts (20 μL) were mixed with 140 μL of sodium phosphate buffer (0.1 mM, pH 8.0) and 20 μL of AChE or BuChE solution (0.28 U/mL in sodium phosphate buffer 0.1 mM, pH 7.0) in 96 well microplates and incubated at RT for 15 min. The reaction was initiated by adding 10 μL of the substrates of the enzymes (acetylthiocholine or butyrylthiocholine iodide, 4 mg/mL diluted in sodium phosphate buffer 0.1 mM, pH 8.0) and with 20 μL of 5,50–dithio–bis (2–nitrobenzoic acid) (DTNB) 1.2 mg/mL in ethanol. The absorbance at 412 nm was read after 10 min of incubation at RT. Galanthamine was used as the positive control at the concentration of 1 mg/mL.

#### 3.4.2. α–Amylase Inhibition

The α–amylase inhibitory activity was determined following Xiao et al. (2006) [60]. Extracts (40 μL) were mixed with 40 μL of amylase solution (100 U/mL in 0.1 M sodium phosphate buffer, pH 7.0) and 40 μL of 0.1% starch solution (diluted in the previous buffer) in a 96-well microplate. After 10 min of incubation at 37 °C, 20 μL of 1 M hydrochloric acid (HCl) and 100 μL of iodide solution (5 mM iodine (I_2_) + 5 mM potassium iodide (KI), in distilled water) were added. The absorbance was measured at 580 nm and acarbose (10 mg/mL) was used as positive control.

#### 3.4.3. α–Glucosidase Inhibition

The extracts were evaluated for inhibition against microbial α–glucosidase (from *Saccharomyces cerevisiae*) following Rodrigues et al., 2015 [61]. Extracts (50 μL) were mixed with 100 μL of the enzyme solution (1.0 U/mL, in 0.1 M sodium phosphate buffer, pH 7.0), and incubated for 10 min at 25 °C. Subsequently, 50 μL of 5 mM p–nitrophenyl–α–d–glucopyranoside (NGP; diluted in 0.1 M sodium phosphate buffer, pH 7.0) were added. Finally, the absorbance was read at 405 nm after 5 min of incubation at 25 °C. Acarbose (10 mg/mL) was used as positive control.

#### 3.4.4. Lipase Inhibition

The inhibitory activity on porcine lipase was evaluated according to McDougall et al., 2009 [62] adapted to 96-well microplates [59]. In brief, extracts (20 μL), were mixed with 200 μL of Tris–HCl buffer (100 mM, pH 8.2), 20 μL of the enzyme solution (1 mg/mL in Tris–HCl buffer), and 20 μL of the substrate (4–nitrophenyl dodecanoate, 5.1 mM in ethanol). After 10 min of incubation at 37 °C, the absorbance was read at 410 nm. Orlistat (1 mg/mL) was used as the positive control.

#### 3.4.5. Tyrosinase Inhibition

The inhibitory activity against tyrosinase was determined following Zengin 2016 [63] with modifications. The extracts (70 μL) were mixed in 96-well microplates with 30 mL of the enzyme (333 units/mL in phosphate buffer, pH 6.5) and incubated at RT for 5 min. Afterwards, 110 μL of the substrate (L–tyrosine, 2 mM in water) were added and incubated for 30 min at RT. The absorbance was measured at 492 nm and arbutin (1 mg/mL) was used as positive control.

### 3.5. UHPLC–HR–MS/MS Profiling of the Extracts

The chemical profiling of the extracts was performed on a Thermo Scientific™ UltiMate™ 3000 UHPLC, equipped with an Orbitrap Elite (Thermo Fisher Scientific, Waltham, MA, USA) mass spectrometer with a Heated Electro–Spray Ionization source (HESI–II; Thermo Scientific). The extracts (5 µL) were diluted in methanol (pure LC–MS grade, 1:10), injected and separated using a Thermo Scientific Accucore RP–18 column (2.1 × 100 mm, 2.6 µm) in a 40 min run following the method described by Silva et al., 2022 [64]. Xcalibur v4.1 Qual Browser (Thermo Scientific, Waltham, MA, USA) was used for LC–MS data acquisition. Thermo “.raw” data files were converted to “.mzXML” format in centroid mode using Proteowizard [65] and imported in MZmine version 3.2.3 [66] for feature finding, alignment and extraction. Feature intensities were assessed as the peak area in the extracted ion chromatogram (XIC). Blank was used for background feature removal and final features were exported as “.mgf” and “.csv” files. Molecular Networking and Spectral Library Search were performed with the Feature-Based Molecular Networking (FBMN) workflow [67] on GNPS (https://gnps.ucsd.edu, accessed on 10 December 2023) [68]. The data was filtered by removing all MS/MS fragment ions within +/− 17 Da of the precursor *m*/*z*. MS/MS spectra were window filtered by choosing only the top 6 fragment ions in the +/− 50 Da window throughout the spectrum. The precursor ion mass tolerance was set to 0.05 Da and the MS/MS fragment ion tolerance to 0.05 Da. A molecular network was then created where edges were filtered to have a cosine score above 0.70 and more than 6 matched peaks. Further, edges between two nodes were kept in the network if and only if each of the nodes appeared in each other’s respective top 10 most similar nodes. Finally, the maximum size of a molecular family was set to 100, and the lowest scoring edges were removed from molecular families until the molecular family size was below this threshold. The spectra in the network were then searched against GNPS spectral libraries [68,69]. The library spectra were filtered in the same manner as the input data. All matches kept between network spectra and library spectra were required to have a score above 0.6 and at least 4 matched peaks. The DEREPLICATOR was used to annotate MS/MS spectra [70]. SIRIUS 5 [71] was used for in silico annotation of features [72] combining three different tools: ZODIAC [73], CSI:FingerID [74], and CANOPUS based on ClassyFire ChemOnt ontology [75,76]. Feature identification levels followed Sumner et al., 2007 [77].

### 3.6. Fatty Acid Methyl Esters Profiling

#### 3.6.1. Lipids Extraction and Transesterification

Direct transesterification of lipids and free fatty acids (FA) to their corresponding fatty acid methyl ester (FAME) followed the protocol reported by Lepage and Roy with modifications [59,78]. In brief, lyophilized algal biomass (100 mg) was mixed with 1.5 mL of the derivatization solution (methanol/acetyl chloride, 20:1, *v*/*v*). After homogenization in an ultrasound–water bath for 30 min, at room temperature (RT), 1 mL of hexane was added, and samples were heated at 100 °C for 60 min. Samples were cooled in an ice bath for 15 min. Finally, 1 mL of distilled water was added, samples were centrifuged for 5 min at 5000× *g*, the supernatants were collected, filtered (0.2 nm) and used for the determination of FAMEs profile. FAMEs extractions were performed in triplicates.

#### 3.6.2. Determination of FAMEs Profile by GC–MS

The FAMEs profiles of diatoms biomass were analyzed using an Agilent GC–MS (Agilent Technologies 6890 Network GC System, 5973 Inert Mass Selective Detector, Agilent Technologies, Wilmington, DE, USA) coupled with a ZB–5MS capillary column (30 m × 0.25 mm internal diameter, 0.25 μm film thickness, Phenomenex, Torrance, CA, USA) using helium as the carrier gas. Briefly, samples were injected at 300 °C, the temperature profile of the GC oven was 60 °C (1 min), 30 °C/min to 120 °C, 4 °C/min to 250 °C, and 20 °C/min to 300 °C (4 min). For the identification of FAMEs, the total ion mode was used and Supelco^®^ 37 Component FAME Mix (Sigma-Aldrich, Sintra, Portugal) was used as standard. Values were expressed as percentages of total FAMEs.

### 3.7. Fucoxanthin Spectrophotometric Quantification

Acetone and methanol extracts were dried under nitrogen and resuspended in ethanol. The absorbance (Ax) was measured at 445, 663 and 750 nm by a UV–visible spectrophotometer (Genesys 50, Thermo Scientific). Fucoxanthin concentrations, expressed as mg g^−1^ dry weight (DW), was estimated using the following equation [79]:Fucoxanthin (mg/g DW) = [(6.39 × A445 − 5.18 × A663 + 0.312 × A750 − 5.27)/W] × V
where:

W = sample weight (g of dry weight)

V = ethanol volume (L)

A455, A663, A750 = absorbance at x nm

### 3.8. Data Presentation and Statistical Analysis

Statistical analyses were performed in R–Statistics^®^ 3.5.3 version. The results of antioxidant and enzymatic assays were analyzed using one-way ANOVA followed by a Tukey’s post hoc tests with multiple comparisons. FAMEs profiles were compared using a *t*-test. When the obtained activities of the extracts tested at the concentration of 10 mg/mL were above 50%, maximal effective concentrations (EC_50_ mg/mL) and the half-maximal concentration values (IC_50_ mg/mL) were calculated by sigmoidal fitting of the data in the GraphPad Prism V 5.0 program (GraphPad Software, La Jolla, CA, USA).

## 4. Conclusions

In this work the Antarctic diatoms *C. ineffabilis* IMA082A and *C. zucchelli* IMA088A were explored for the first time as a potential source of bioactive products for nutraceutical, biotechnological and pharmaceutical industries. Our results suggest that both species might represent sources of antioxidants, specifically the acetone extract from *C. ineffabilis* which displayed a strong capacity to chelate copper. High to moderate inhibitory activities towards cholinesterase enzymes, α–amylase and lipase were reported for both species, thus suggesting that further work should be carried out aiming to explore its possible application in the treatments of neurodegenerative diseases and in the management of T2DM, obesity and hyperlipidemia. The FAMEs profiles of *C. ineffabilis* and *C. zucchelli* were characterized by a high proportion of PUFAs, which are desirable molecules for the nutraceutical industry and aquaculture. The trajectory of future studies might aim to evaluate the pigment profile and the phenolic profile of these diatoms to better understand their biological activities. Moreover, different growth conditions, such as nitrogen and light limitations, could be further tested to maximize the production of different classes of bioactive metabolites. Finally, we would like to highlight the importance of studying Antarctic photosynthetic organisms as source of bioactive molecules not only for their economic and social importance, but also for their ecological value. Increasing our effort in the study of these poorly known species would help us to identify potential areas for species management and conservation in Antarctica.

## Figures and Tables

**Figure 1 marinedrugs-22-00035-f001:**
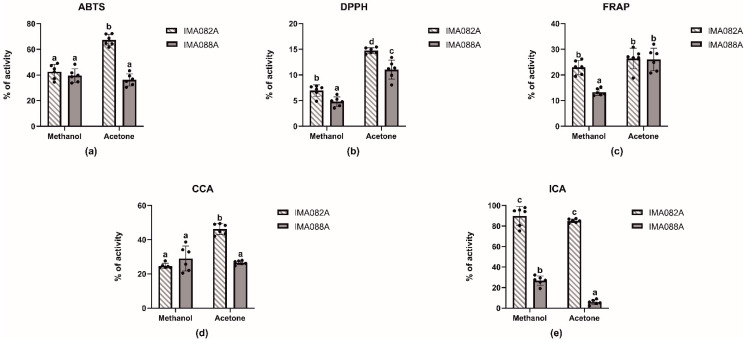
Antioxidant activities of the acetone and methanol extracts *of C. ineffabilis* IMA082A and *C. zucchelli* IMA088A. Radical scavenging activity on ABTS•+ (**a**) and DPPH• (**b**) radicals, FRAP (**c**) CCA (**d**) and ICA (**e**). Results are expressed as antioxidant activity (% of activity) at the concentration of 10 mg/mL. The letters above the bars in the bar charts indicate significantly different groups (Multiple Comparisons of Means: Tukey’s HSD, 95% family-wise confidence level).

**Figure 2 marinedrugs-22-00035-f002:**
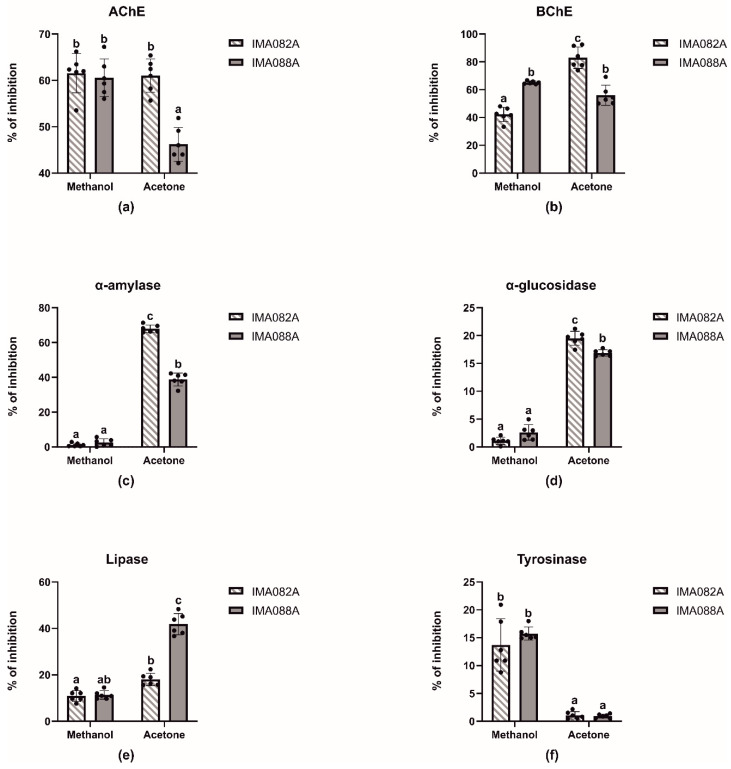
Enzymatic inhibitory properties of the acetone and methanol extracts *C. ineffabilis* IMA082A and *C. zucchelli* IMA088A. Results are expressed as inhibitory activity (% of inhibition) at the concentration of 10 mg/mL. The letters above the bars in the bar charts indicate significantly different groups (Multiple Comparisons of Means: Tukey’s HSD, 95% family-wise confidence level).

**Figure 3 marinedrugs-22-00035-f003:**
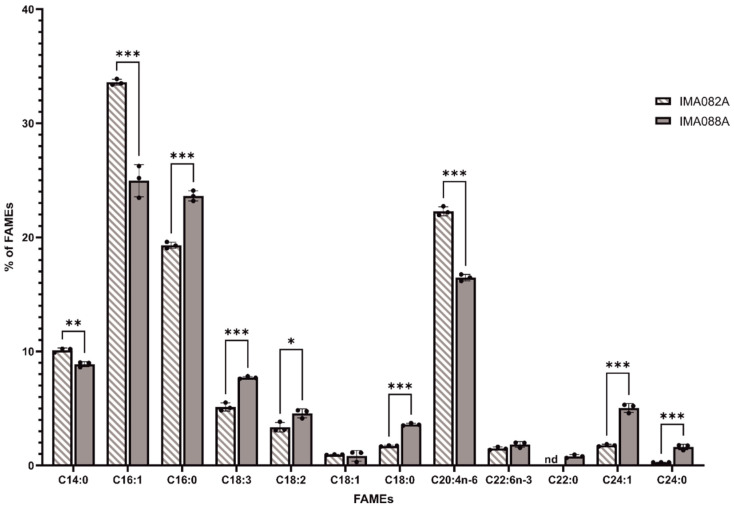
FAMEs profiles of *C. ineffabilis* IMA082A and *C. zucchelli* IMA088A. Results are expressed as percentage of total FAMEs. Asterisks indicate statistically significant differences (‘***’ = *p*–value < 0.001; ‘**’ = *p*–value < 0.01 and ‘*’ = *p*–value < 0.05).

**Table 1 marinedrugs-22-00035-t001:** Half maximal effective concentrations (EC_50_, mg/mL) for extracts displaying an activity above 50% when tested at the concentration of 10 mg/mL. Values represent the mean ± standard error of mean (SEM) performed six times (*n* = 6). For the same column, different letters indicate significant differences (Multiple Comparisons of Means: Tukey Contrast, 95% family-wise confidence level).

Species	Extract	ABTS	ICA
*C. ineffabilis* IMA082A	Acetone 80%	6.79 ± 0.21	5.73 ± 0.53 b
	Methanol 50%	–	2.06 ± 0.70 a
*C. zucchelli* IMA088A	Acetone 80%	–	–
	Methanol 50%	–	–

**Table 2 marinedrugs-22-00035-t002:** Half maximal inhibitory concentration (IC_50_, mg/mL) for extracts displaying an activity above 50% when tested at the concentration of 10 mg/mL. Values represent the mean ± standard error of mean (SEM) performed six times (*n* = 6). For the same column, different letters indicate significant differences (Multiple Comparisons of Means: Tukey Contrast, 95% family-wise confidence level).

Species	Extract	AChE	BChE	α-Amylase
***C. ineffabilis* IMA082A**	Acetone 80%	7.99 ± 2.62 a	1.81 ± 0.46 a	6.87 ± 0.17
	Methanol 50%	6.03 ± 1.34 a	–	–
***C. zucchelli* IMA088A**	Acetone 80%	–	4.34 ± 0.39 b	–
	Methanol 50%	9.70 ± 1.44 a	6.59 ± 0.35 c	–

**Table 3 marinedrugs-22-00035-t003:** Putative annotated compounds (level 2) in acetone extracts of *C. ineffabilis* IMA082A and *C. zucchelli* IMA088A. 🗸 = compound presence.

Peank No.	*m*/*z* Value	RT (min)	Adduct	Annotated Molecular Formula	Annotated Compound	IMA082A	IMA088A
1	197.1172	13.3438	[M + H]+	C11H16O3	Loliolide	🗸	🗸
2	323.221	22.055	[M + H]+	C19H30O4	MG(16:4)	🗸	🗸
4	213.1632	22.3055	[M − H_4_O_2_ + H]+	C16H24O2	6,9,12,15-Hexadecatetraenoic acid	🗸	🗸
5	327.252	22.4687	[M − H_4_O_2_ + H]+	C19H34O4	MG(16:2)	🗸	🗸
6	337.2362	22.4864	[M − H_4_O_2_ + H]+	C20H34O5	Prostaglandin D1	🗸	
7	299.1996	22.6948	[M − H_4_O_2_ + H]+	C20H30O4	Resolvin E3	🗸	
8	321.2416	22.8543	[M − H_4_O_2_ + H]+	C20H34O4	11,12-Dheta	🗸	🗸
9	325.2362	22.9608	[M + H]+	C19H32O4	MG(16:3)	🗸	🗸
10	509.2702	22.9608	[M + Na]+	C25H42O9	MGMG(16:3)	🗸	🗸
11	303.231	23.0676	[M + H]+	C20H30O2	Eicosapentaenoic acid	🗸	
12	530.331	23.6646	[M + H_3_N + H]+	C27H44O9	MGMG(18:4)	🗸	🗸
13	351.2522	23.6919	[M + H]+	C21H34O4	MG(18:4)	🗸	🗸
14	516.307	23.8108	[M + H]+	C26H46NO7P	LPC(18:4)	🗸	🗸
15	465.26	23.9435	[M + H]+	C22H43O9P	LPG(16:1)		🗸
16	542.3229	24.7478	[M + H]+	C28H48NO7P	LPC(20:5)	🗸	🗸
17	508.3471	24.7865	[M + H_3_N + H]+	C25H46O9	MGMG(16:1)	🗸	🗸
18	518.323	24.9074	[M + H]+	C26H48NO7P	LPC(18:3)	🗸	🗸
19	301.2157	25.0337	[M − H_4_O_2_ + H]+	C20H30O3	11-HEPE	🗸	🗸
20	574.3244	25.2321	[M + H_3_N + H]+	C25H48O11S	SQMG(16:0)	🗸	🗸
21	494.3219	25.4718	[M + H]+	C24H48NO7P	LPC(16:1)	🗸	🗸
22	505.2522	25.6538	[M + Na]+	C22H43O9P	LPG(16:1)	🗸	🗸
23	447.2491	25.6819	[M − H_4_O_2_ + H]+	C22H43O9P	PG(16:1)	🗸	🗸
24	544.3382	25.7362	[M + H]+	C28H50NO7P	LPC(20:4)	🗸	🗸
25	568.3384	25.789	[M + H]+	C30H50NO7P	LPC(22:6)	🗸	🗸
26	520.3636	26.2199	[M + H]+	C26H50NO7P	LPC(18:2)	🗸	🗸
27	449.2647	26.357	[M − H_4_O_2_ + H]+	C22H45O9P	PG(16:0)	🗸	🗸
28	296.2577	26.6736	[M + H_3_N + H]+	C18H30O2	alpha-Linolenic acid	🗸	🗸
29	522.3537	28.4701	[M + H]+	C26H52NO7P	LPC(18:1)	🗸	🗸
30	496.3379	28.6544	[M + H]+	C24H50NO7P	LPC(16:0)	🗸	🗸
31	609.2693	30.5862	[M + H]+	C35H36N4O6	10-Hydroxyphaeophorbide	🗸	🗸

**Table 4 marinedrugs-22-00035-t004:** Putative annotated compounds (level 2) in methanol extracts of *C. ineffabilis* IMA082A and *C. zucchelli* IMA088A. 🗸 = compound presence.

Peank No.	*m*/*z* Value	RT (min)	Adduct	Annotated Molecular Formula	Annotated Compound	IMA082A	IMA088A
1	148.0598	0.9564	[M + H]+	C5H9NO4	Glutamate	🗸	🗸
2	132.1012	1.6404	[M + H]+	C6H13NO2	Leucine	🗸	🗸
3	197.1162	13.3722	[M + H]+	C11H16O3	Loliolide	🗸	🗸
4	507.2532	22.3166	[M + Na]+	C25H40O9	MGMG(16:4)		🗸
5	274.2734	22.4747	[M + H]+	C16H35NO2	Lauryldiethanolamine	🗸	🗸
6	318.2998	22.6623	[M + H]+	C18H39NO3	Phytosphingosine	🗸	🗸
7	290.2684	22.8782	[M + H]+	C16H35NO3	Hexadecaphytosphingosine	🗸	🗸
8	530.3297	23.6846	[M + H_3_N + H]+	C27H44O9	MGMG(18:4)		🗸
9	516.3079	23.7679	[M + H]+	C26H46NO7P	LPC(18:4)	🗸	🗸

**Table 5 marinedrugs-22-00035-t005:** Fucoxanthin content of the acetone and methanol extracts of *C. ineffabilis* IMA082A and *C. zucchelli* IMA088A. The results are expressed as mg/g of dry weight (DW). Different letters indicate significant differences (Multiple Comparisons of Means: Tukey Contrast, 95% family-wise confidence level). Values represent the mean ± standard error of mean (SEM) performed three times (*n* = 3).

Species	Extract	Fucoxanthin (mg/g DW)
*C. ineffabilis* IMA082A	Acetone 80%	0.19 ± 0.04 ab
	Methanol 50%	0.25 ± 0.02 bc
*C. zucchelli* IMA088A	Acetone 80%	0.14 ± 0.03 a
	Methanol 50%	0.32 ± 0.05 c

## Data Availability

The dataset is available upon request from the corresponding author.

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
