# Peer review of "Bioprospection of the Antarctic Diatoms *Craspedostauros ineffabilis* IMA082A and *Craspedostauros zucchelli* IMA088A"

_marinedrugs, 2024, doi:10.3390/md22010035_

Round 1

Reviewer 1 Report

Comments and Suggestions for Authors

The systematic exploration of microorganisms in polar regions for valuable products holds significant potential for commercialization and societal advancement. This study represents a noteworthy effort in uncovering novel diatom species for potential biotechnological applications. While the overall work shows promise, there are specific areas that require attention before considering the manuscript for publication.

1.     The manuscript focuses on the bioprospection of Antarctic diatoms but lacks illustrations of Diatom resources and Diatom taxonomy.

2.     In comparison to other microalgae, what novel compounds specifically exist in Craspedostauros ineffabilis IMA082A and Craspedostauros zucchelli IMA088A?

3.     The low temperature and high intensity of ultraviolet radiation are characteristic features of Antarctica. Do these environmental factors contribute to the chemical compositions of these new species?

4.     It is recommended to include biomass and/or microscopic images of IMA082A and IMA088A in the manuscript to provide readers with an intuitive understanding of these new polar species.

Comments on the Quality of English Language

Moderate editing of English language required

Author Response

Dear Reviewer 1,

We have reorganized the structure of the manuscript into the following sections, Introduction, Results and Discussion, Materials and Methods, and Conclusions, as required by the editor. We have incorporated your suggestions as well as those from Reviewer 2. Below, we address your main comments:

1) The manuscript focuses on the bioprospection of Antarctic diatoms but lacks illustrations of Diatom resources and Diatom taxonomy.

Answer 1) We described Craspedostauros ineffabilis IMA082A and Craspedostauros zucchelli IMA088A as two new species in a previous publication entitled: ‘Molecular, Morphological and Chemical Diversity of Two New Species of Antarctic Diatoms, Craspedostauros ineffabilis sp. nov. and Craspedostauros zucchellii sp. nov.’ J. Mar. Sci. Eng. 2022, 10(11), 1656; https://doi.org/10.3390/jmse10111656. This work extensively delved into the light and scanning electron microscope observations of strains IMA082A and IMA088A, DNA sequence data, and a comprehensive phylogenetic analysis.

Given that the systematic placement of both Antarctic strains was rigorously established using a modern integrative approach merely a year ago, we consider that further taxonomic insights are not the primary focus of our current study. Instead, our present investigation centered on evaluating the biological activity of C. ineffabilis and C. zucchellii extracts, alongside their chemical profiling and lipid profile analysis.

2) In comparison to other microalgae, what novel compounds specifically exist in Craspedostauros ineffabilis IMA082A and Craspedostauros zucchelli IMA088A?

Answer 2) This is a very good point. As reported in Table 3 and 4 (Results section), the compound identified in both species included fatty acids, acylglycerols, galactosylglycerols, phosphatidylglycerols, lysophosphatidylcholines and derivatives with elevated degree of desaturation in their carbon chain. This is a common strategy of Antarctic species to cope with cold temperatures. The increasing percentage of unsaturated fatty acids (and their derivatives) help to increase membranes fluidity, which is fundamental for survival at freezing temperatures (lines 277-283). Other compound detected by UHPLC were carotenoid derivatives, such as loliolide, chlorophyll derivatives, such as 10-Hydroxyphaeophorbide, and amino acids, such as glutamate and leucine. All these compounds are common algal metabolites and some of them were already investigated for their bioactive properties, as reported in the discussion section of the present manuscript. Whether fucoxanthin was only detected with a spectrophotometric method, we expect that future studies targeting xanthophylls will provide new insights on the pigment profile of C. ineffabilis and C. zucchelli.

Our metabolomic strategy was untargeted, thus it was not possible to describe new compounds from these two species since our spectral data were compared with those available in public libraries. However, future targeted analysis aiming to describe the chemical structure of apolar compounds from the Antarctic strain could reveal novel molecules, particularly of the class of oxylipins, as reported in Table 3, which are recently described compounds from diatoms, whose roles are currently poorly studied.

3) The low temperature and high intensity of ultraviolet radiation are characteristic features of Antarctica. Do these environmental factors contribute to the chemical compositions of these new species?

Answer 3) This is another good point. These species are adapted to harsh environmental conditions (lines 48-55) and developed unique strategies to cope with these stressors. A future of our research will be to test the response of C. ineffabilis and C. zucchelli to some stressors (i.e. UV, light intensity, temperature, salinity, …) and to evaluate the changes in their chemical composition.

4)  It is recommended to include biomass and/or microscopic images of IMA082A and IMA088A in the manuscript to provide readers with an intuitive understanding of these new polar species.

Answer 4) As previously mentioned (see Answer 1), we encourage readers to explore the systematic details of these two diatoms in the publication referenced earlier. These details are also outlined in the introduction of the manuscript (line 55-66).

We sincerely appreciate your time, attention, and invaluable contributions that significantly enriched the quality and comprehensiveness of our work.

Reviewer 2 Report

Comments and Suggestions for Authors

The study conducted by Trentin et al. is interesting, novel, and well-written. I suggest minor revision before further processing.

Comments:

Line 82: define these abbreviations and all abbreviations in the methodology.

L150: α-amylase

L158: α–glucosidase

L159: rewrite

L180: UHPLC.

Please add the p-value to the results.

L304: UHPLC–HR–MS

L303: PLEASE add the concentration or percentage (area %) of each compound detected for both extracts.

L400-404: rewrite and adjust the writing of the references. I suggest splitting it into two sentences.

L404-408: split into two sentences. Define T2DM.

The authors should revise the writing style of references in the discussion section.

L426-428: rephrase.

L430-431: rephrase.

Table S1: the authors didn’t mention the results compared to the conventional antioxidants (BHT and EDTA) in the text. The same for Table S2. The authors should compare the results with the control groups and show them in the text not only the suppl. Tables.

Comments on the Quality of English Language

Moderate editing is required.

Author Response

Dear Reviewer 2,

We have reorganized the structure of the manuscript into the following sections, Introduction, Results and Discussion, Materials and Methods, and Conclusions, as required by the editor. We have incorporated your suggestions as well as those from Reviewer 1. Below, we address your main comments:

Comments:

1) Line 82: define these abbreviationsand all abbreviations in the methodology.

Answer 1) We inserted all the abbreviations, as required. (see lines 293-294)

2) L150: α-amylase

Answer 2) We corrected the misspelled word.

3) L158: α–glucosidase

Answer 3) We corrected the misspelled word, thanks for noting that.

4) L159: rewrite

Answer 4)

5) L180: UHPLC

Answer 5) Correction made.

6) Please add the p-value to the results.

Answer 6) Instead of reporting the p-values, we presented significant differences resulting from ANOVA post-hoc tests (Tukey test) using different letters, as reported in Figures 1 and 2 and in Tables 1 and 2. This make it easier to the reader to highlight significant differences. In Figure 3, Asterisks indicate statistically significant differences (‘***’ = p–value < 0.001; '**' = p–value < 0.01 and '*' = p–value < 0.05).

7) L304: UHPLC–HR–MS

Answer 7) Correction made.

8) L303: PLEASE add the concentration or percentage (area %) of each compound detected for both extracts.

Answer 8) Since we performed a screening of the extracts following an untargeted approach, we could only detect the presence or the absence of a metabolite in our UHPLC–HR–MS/MS runs. The identification of FAMEs, instead, was made using commercial standards, thus the main FAMEs were identified and their abundance was reported as percentage. However, you highlighted a good point that we will explore in the future. The determination of the concentrations of interesting metabolites, such as oxylipins, and their specific activity will be addressed soon.

9) L400-404: rewrite and adjust the writing of the references. I suggest splitting it into two sentences

Answer 9) Thanks for the suggestion. We divided the sentence in two shorter sentences and refreshed them (see lines 236-240).

10) L404-408: split into two sentences. Define T2DM.

Answer 10) We have rephrased the sentence as recommended, splitting it for improved clarity. Additionally, to address the request for clarity, T2DM refers to Type–2 Diabetes Mellitus, a chronic metabolic condition characterized by insulin resistance and high blood sugar levels. (see lines 240-244).

11) The authors should revise the writing style of references in the discussion section.

Answer 11) We checked all the references again.

12) L426-428: rephrase.

Answer 12) We have opted to exclude the sentence as it was not contributing to the discussion.

13) L430-431: rephrase.

Answer 13) We modified the sentence as required (see lines 264-266).

14) Table S1: the authors didn’t mention the results compared to the conventional antioxidants (BHT and EDTA) in the text. The same for Table S2. The authors should compare the results with the control groups and show them in the text not only the suppl. Tables.

Answer 14) We decided to keep the results of the positive controls for the antioxidants and for the enzymatic assays in the supplementary materials to enhance the readability of the main text and the impact of the figures presented in our study. However, all data are available in the supplementary materials which can be download easily by other researchers.

We sincerely appreciate your time, attention, and invaluable contributions that significantly enriched the quality and comprehensiveness of our work.

Round 2

Reviewer 1 Report

Comments and Suggestions for Authors

The author has responded to the reviewer's inquiries. I recommend considering the manuscript for publication in this journal.

Comments on the Quality of English Language

Minor editing of English language required